# Examining the effectiveness of consuming flour made from agronomically biofortified wheat (Zincol-2016/NR-421) for improving Zn status in women in a low-resource setting in Pakistan: study protocol for a randomised, double-blind, controlled cross-over trial (BiZiFED)

Nicola M Lowe,[1] Muhammad Jaffar Khan,[2] Martin R Broadley,[3] Munir H Zia,[4] Harry J McArdle,[3] Edward J M Joy,[5] Heather Ohly,[1] Babar Shahzad,[2] Ubaid Ullah,[2] Gul Kabana,[6] Rashid Medhi,[6] Mukhtiar Zaman Afridi[7]

For numbered affiliations see end of article.

**Correspondence to**
Professor Nicola M Lowe; NMLowe@uclan.ac.uk

## ABSTRACT

**Introduction** Dietary zinc (Zn) deficiency is a global problem, particularly in low-income and middle-income countries where access to rich, animal-source foods of Zn is limited due to poverty. In Pakistan, Zn deficiency affects over 40% of the adult female population, resulting in suboptimal immune status and increased likelihood of complications during pregnancy.

**Methods and analysis** We are conducting a double-blind, randomised controlled feeding study with cross-over design in a low-resource setting in Pakistan. Households were provided with flour milled from genetically and agronomically biofortified grain (Zincol-2016/NR-421) or control grain (Galaxy-2013). Fifty households were recruited. Each household included a woman aged 16–49 years who is neither pregnant nor breastfeeding, and not currently consuming nutritional supplements. These women were the primary study participants. All households were provided with control flour for an initial 2-week baseline period, followed by an 8-week intervention period where 25 households receive biofortified flour (group A) and 25 households receive control flour (group B). After this 8-week period, groups A and B crossed over, receiving control and biofortified flour respectively for 8 weeks. Tissue (blood, hair and nails) have been collected from the women at five time points: baseline, middle and end of period 1, and middle and end of period 2.

**Ethics and dissemination** Ethical approval was granted from the lead university (reference no. STEMH 697 FR) and the collaborating institution in Pakistan. The final study methods (including any modifications) will be published in peer-reviewed journals, alongside the study outcomes on completion of the data analysis. In addition, findings will be disseminated to the scientific community via conference presentations and abstracts and communicated to

### Strengths and limitations of the study

► The study is the first randomised controlled trial to determine the effectiveness of consuming flour from the Zincol-2016/NR-421 strain of wheat on Zn status in women of childbearing age.
► The study will evaluate the use of novel biomarkers of Zn status in low-resource settings.
► A limitation of this study is the relatively small sample size (n=50). This is compensated for by the cross-over design, which strengthens the statistical power.

the study participants through the village elders at an appropriate community forum.

**Registration details** The trial has been registered with the ISRCTN registry, study ID ISRCTN83678069.

## INTRODUCTION

Dietary zinc (Zn) deficiency is a global problem, affecting 17% of the world's population, with the greatest burden in low-income and middle-income countries.[1 2] The most recent national survey in Pakistan indicates that over 40% of women are Zn deficient.[3] The consequences of Zn deficiency are profound and far-reaching, ranging from increased individual morbidity and mortality to problems with community and regional economic development.[4]

Attempts to alleviate the problem of Zn deficiency, both in Pakistan and in other

countries, have followed several strategies. Providing micronutrient supplements, usually using 'sprinkles', has been attempted under different circumstances, generally with some success.[5] However, it is an expensive and intensive approach, and without close supervision it can often be that the intended target, for example a pregnant woman, does not end up receiving the supplement. Fortification of foods at the processing stage has also been widely adopted, but there are significant problems with compliance and efficacy, both at the fortification stage where quality control is ineffective and at the level of consumer uptake. A major issue is that poorer households in rural areas of developing countries typically grow the majority of their food and process it at village or community level. Thus, cereal fortification at large-scale processing facilities may not reach poorer households, where micronutrient deficiencies are often more prevalent and severe.

An alternative approach is to intervene at agricultural production stage through biofortification to achieve greater concentrations of bioavailable vitamins or elements in the edible portion of crops.[6] Biofortification can be realised through crop breeding or application of element-enriched fertilisers, known as 'agronomic biofortification', which is particularly important in Pakistan where plant-available soil Zn concentrations are low.[7] Such Zn-deficient soils that constitute 52% at national scale are also prevalent in Peshawar, Khyber Pakhtunkhwa Province, where plant-available soil Zn concentrations are significantly below the threshold 1 mg/kg in 75% of the soils analysed. Farmers in Pakistan rarely apply Zn fertilisers to their crops except for rice. Biofortification approaches have focused on staple crops because they are widely consumed including households with low purchasing power. Recently, varieties of maize, pearl millet and beans with high iron (Fe) and Zn concentrations have been developed by the HarvestPlus programme for deployment in low-income settings. Human feeding studies have demonstrated that such varieties can lead to improvements in Fe and Zn intake and status,[8 9] although there may be issues with low bioavailability.[10] Agronomic biofortification has not previously been deployed in a low-income setting but does have policy precedent in Finland, where granular fertilisers were enriched with selenium (Se) since the mid-1980s with consequent increases in human dietary Se intakes and eventual elimination of Se deficiency as a public health problem.[11]

Preliminary studies have shown that Zn concentrations in wheat can be raised substantially through biofortification strategies.[12] It is notable that increasing levels of nutrients in the whole plant is not sufficient; elements must be translocated to the edible component of the grain (endosperm) and must be bioavailable. Phytic acid (PA) is the most important inhibitor of Zn absorption from wheat; therefore, the Zn:PA molar ratio of the endosperm is an important consideration. The process of element accumulation in the endosperm is not simple. For example, Zn moves through many different transport processes prior to being deposited in the seed.[13] In a recent set of experiments within the HarvestPlus programme over a period of 8 years, scientists in Pakistan using traditional breeding techniques have been able to generate a 'high Zn' variety of wheat that is adapted to Pakistan conditions. The variety (Zincol-2016/NR-421) contains more Zn per kilogram of grain compared with traditional varieties. However, grain Zn concentration will also depend on the availability of Zn in the soil, and there are likely to be synergistic effects of breeding for high Zn with application of Zn fertilisers.

The efficacy of increasing dietary intake through biofortification on individual and population Zn status is notoriously difficult to measure. The assessment of Zn status has been the subject of considerable research and is summarised comprehensively in two recent reviews supported by multinational expert groups.[14 15] Both reviews conclude that plasma or serum Zn concentration, the most widely used biomarker, is useful provided that various confounding factors are considered (including concurrent infection and inflammation). However, the plasma/serum must be collected following stringent methodology to avoid contamination and can be difficult to achieve in field settings. Hair Zn concentration is also potentially useful, and samples are easy to collect and store. In addition, emerging techniques, including the measurement of DNA fragmentation using the Comet assay and analysis of nail Zn concentration using laser ablation, are gaining credibility as useful biomarkers of Zn status. However, field trials are needed to evaluate their usefulness.

The primary aim of this study is to measure the impact of consuming flour made from the recently released high-Zn wheat grain variety Zincol-2016/NR-421 on dietary Zn intake and biomarkers of Zn status in a low-resource rural community setting in Pakistan. The secondary aim is to evaluate the potential usefulness of new biomarkers of Zn status. These aims will be achieved through conducting a randomised controlled feeding study with a cross-over design. Comparison of outcome measures will be made following consumption of flour made from the biofortified wheat variety, Zincol-2016/NR-421, and the control, non-biofortified variety, Galaxy. The trial has been registered with the ISRCTN registry, study ID ISRCTN83678069.[16]

## METHODS AND ANALYSIS
### Study setting and recruitment
This study was conducted from the Health Centre that serves a community of approximately 5000 households in an area of 8 km² on the outskirts of Peshawar in Khyber Pakhtunkhwa (KPK). This community was chosen because previous studies have indicated that Zn deficiency is widespread, the diet is limited and primarily vegetable-based (thus low in bioavailable Zn) and the staple is wheat which is used for making chapati and roti.[17] The catchment area, known as a Union Council,

is made up of 10 villages (clusters). Out of 10 clusters in the union council, 5 were randomly selected, from which equal numbers of households were randomly selected for the study. Selection of these clusters and households into intervention or control arm was done using a block design. Water samples were collected in duplicate from four different communal tube wells (hand pump) supplying each village and analysed for mineral content by inductively coupled plasma-mass spectrometry (ICP-MS). The mean Zn concentration of the water from one cluster was high enough to significantly impact on daily Zn intake (>250 µg/L); therefore, this cluster was eliminated from the study and an alternative cluster was randomly selected. The mean Zn concentration in the water analysed from the five included clusters was ≤120 µg/L and therefore not considered to add significantly to the daily Zn intake, based on a consumption of 2–4 L water per day (<0.5 mg Zn per day).

Ten households from each cluster were randomly selected. The head of the household was approached, the purpose of the study explained and, if eligible, the family invited to participate in the trial. If the head of the household declined, another house from the cluster was randomly selected and the invitation process repeated. Inclusion criteria were the household included a woman aged 16–49 years who was neither pregnant nor breastfeeding, and not currently consuming nutritional supplements. When there was more than one eligible woman in the household, one was conveniently selected with agreement from the head of the household. There were no additional exclusion criteria. Following informed consent, each household was assigned to treatment arm A or B (refer to figure 1) using a block design. A sample size of 40, 95% significance level (two-sided) and 90% power, will enable an increase in plasma Zn concentration of 3.1 µg/dL to be detected. The target was to recruit 50 households (50 eligible women) to be able to accommodate a 20% attrition rate during the 18-week study. The project manager led the recruitment process because he is known to the community through previous health-related projects associated with the Abaseen Foundation PK (AFPK), a local non-governmental organisation and key member of this project team, along with its sister organisation Abaseen Foundation UK. There is a high level of trust and cooperation between community members and the AFPK who operate the health centre that serves this Union Council; therefore, high levels of participant enrolment has been achieved. The project manager was responsible for allocating participants to each arm of the cross-over trial using a block design. Only the project manager had access to this information.

The participants and the remaining research team members, including the PI and Co-Is, database manager and data analysts were blind to the order in which households received the biofortified or control flour. Unblinding was permissible if any adverse reactions are reported by the households during the trial.

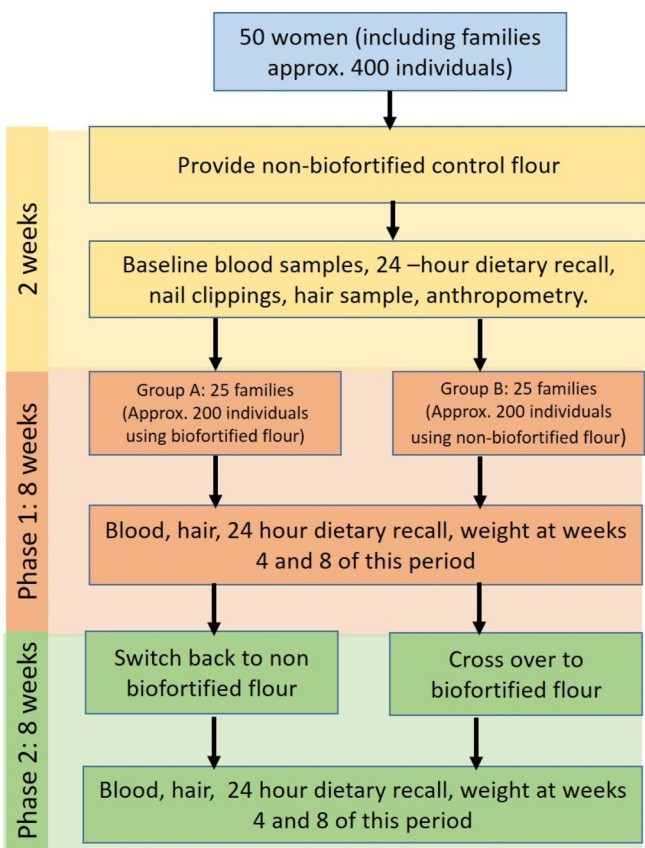

**Figure 1** Schematic overview of objective 1, randomised controlled feeding study with cross-over design.

## Study design

We have conducted a randomised controlled feeding study with cross-over design in which households were provided with flour milled from genetically and agronomically biofortified grain (Zincol-2016/NR-421) or control grain (Galaxy-2013). Galaxy-2013 was chosen for the control grain because it is widely cultivated across the country and is popular among farmers due to its higher yield. It did not receive Zn-containing fertilisers. Zincol-2016 (formerly known as NR-421 strain/line) was bred by the National Agriculture Research System, Pakistan with the support of HarvestPlus and approved by the Punjab Seed Council, Pakistan in 2016.[18] The grain for both the genotypes was sown at a farm in District Vehari , Punjab, Pakistan in November 2016 and harvested in May 2017 by our project partner, Fauji Fertilizer Company (FFC) Limited. During the growing period, Zn-enriched fertiliser (Zn 13% as EDTA Zn) was applied to Zincol-2016 genotype at 1.25 kg/ha through injection into the irrigation system. This was followed by five foliar sprays of Zn ($ZnSO_4 \cdot H_2O$ 33% Zn) applied at 1 kg/ha of product dissolved in 250 L of water and sprayed per hectare at the booting stage. Analyses revealed that the pre-harvest Zn concentration in grains were 62 (SD 7.7) mg/kg and 27 (SD 4.3) mg/kg for the biofortified and control grain, respectively. The grain was transported from the farm to the study field site in clearly labelled sacks (control and biofortified). The bags of grain were manually cleaned

(hand removal of debris, stones, stalks), mixed and milled at a local facility. Care was taken to avoid cross-contamination from previously milled grain. Samples of grain from each sack (top, middle and bottom) were retained for analysis.

The intervention trial was composed of a 2-week baseline equilibration period and two 8-week intervention/control periods. The baseline period started on 1 October 2017. During the baseline period, all the participating households were provided with control flour. After 2 weeks, group A received biofortified flour and group B received control flour. After 8 weeks, the two groups will cross over, with group A receiving control flour and group B receiving biofortified flour for the final 8 weeks (figure 1). Freshly milled flour was provided on a fortnightly basis for consumption by the entire household (average seven individuals, including children). Families will be asked not to consume any other flour within the household during the 18-week study period. Compliance was monitored at each flour distribution time point through interview with the head of the household. The number of meals consumed outside the household by the primary study participant (woman of childbearing age) was recorded. Our previous knowledge of this community indicates that, other than community events such as weddings, it is unusual for women to eat outside the home.

### Primary outcome measures
Flour is used primarily for making roti and chapati in this community. The contribution of the total Zn intake from foods made from flour will be determined using 24-hour recall, coupled with food composition databases (eg, WinDiets 2005 Research software, Robert Gordon University, UK). In addition, a selection of composite meal samples will be analysed. The impact of consuming the biofortified flour on the Zn status of women of childbearing age will be assessed by measuring plasma Zn concentration on ICP-MS analysis, DNA fragmentation by Comet assay, and hair and nail Zn concentration by ICP-MS. In addition, we will measure concentrations of inflammatory biomarkers C-reactive protein (CRP) and alpha 1-acid glycoprotein that will be used to adjust plasma Zn concentration if required.[19]

### Dietary analysis
Dietary nutrient intake will be assessed by analysing the 24-hour recalls that were collected by interviewing the women participants by the study nutritionist, who has local knowledge of the foods consumed and speaks Pushto, the local language. From our previous research,[17] we know that the local diet is very limited in variety and monotonous and therefore a total of five recall interviews, taken at each of the sampling points (figure 1), will give a good estimate of typical nutrient intake. Portion sizes will be estimated using household measures, using items that are familiar to the households (eg, plates, cups, spoons). Macronutrient and micronutrient composition, plus phytate will be calculated from appropriate food composition databases. In addition, a selection of commonly consumed composite meals will be analysed for mineral, macronutrient and phytic acid concentration. During this interview, the nutritionist also collected data regarding adverse health-related events since the last measurement time point. Mothers were also asked to report on the frequency and duration of diarrhoeal episodes among children in the household. Any major concerns were reported to the Health Centre Doctor for follow-up.

### Biomarkers of Zn status
Tissue samples were collected from participants attending the Health Centre at five time points: baseline, middle and end of intervention phase 1, and middle and end of phase 2 (figure 1).

### Plasma Zn concentration
Whole blood was collected into trace element–free tubes (BD Diagnostics, Switzerland) and the plasma separated by centrifugation within 40 min. Plasma 300 µL aliquots were transferred into Eppendorf tubes and stored at −20°C. Plasma samples will be shipped on dry ice to the University of Nottingham for analysis by ICP-MS.

### DNA fragmentation analysis
DNA fragmentation (measured using the Comet assay) has been shown to decrease following dietary Zn supplementation.[20] Whole blood was collected by venous blood draw. Duplicate samples of 50 µL of whole blood was mixed with 50 µL cryopreservation buffer (85% RPMI, 15% DMSO) in a cryovial and stored in a cryopreservation container in a fridge for up to 8 hours. Samples were transferred from the health centre to Khyber Medical University (KMU) on ice and placed in a −80°C freezer overnight (up to 12 hours), then transferred to liquid nitrogen for storage until samples at all five time points have been collected. DNA fragmentation will be quantified using the method of Singh *et al*,[21 22] and Comet Assay IV Lite software (Perceptive Instruments, London, UK) for tail moment measurement.

### Nail and hair analysis
Whole hairs were collected by combing through the hair and collecting individual hairs from the comb, making sure that the tag (cells from follicle) is attached to the proximal end of the hair. Nail samples were collected at baseline. Nail samples will be collected two more times at approximately 7 months (average time taken for a nail to grow from formation in the nail bed to the tip of the finger) from the start of phase 1 and phase 2. The Zn concentration of hair and nail samples will be determined using laser-induced breakdown spectroscopy[23] at NeurobioTex (Galveston, Texas, USA).

### Additional outcomes
### Anthropometry
Height and weight of the women were measured at baseline, and weight was monitored at each sample collection

time point. Height was measured using a height board. The participant was to stand still with her back towards the scale, with the heels, back of the calves, upper back and the back of their head touching the board with the scale. The head was positioned so that the Frankfurt line was straight. The measuring device was lowered gently on the hair until it rested gently on the scalp. For the measurement of weight, women were asked to relax for 5 min before being asked to remove added clothes, shoes, socks, anything in the pocket, watch or necklace. The woman was asked to stand on the marked position on the scale until the digital reading or the reading needle stabilised. For consistency, weight was measured three times and the average calculated. In addition, the height and weight of at least one child under 10 years in each household will be monitored throughout the intervention period. Mothers were also be asked to keep a diary of the frequency and duration of diarrhoeal episodes of children in the household throughout the intervention period.

### Haematological and micronutrient analyses

Haematocrit, haemoglobin, mean corpuscular volume, total iron binding capacity and transferrin saturation will be estimated using clinical haematology methods. Blood samples will be analysed for micronutrient status including serum transferrin receptor and ferritin (sandwich ELISA), Fe and SE concentrations (ICP-MS), vitamin A (retinol-binding protein, RBP) and inflammatory markers ($\alpha$1-acid glycoprotein and CRP).

### Dark adaptation

The potential impact of increasing dietary Zn intake on dark adaptation was measured at baseline, week 2, week 10 and week 18 during the participants' visits to the health centre. Dark adaptation was measured using a novel device developed by Labrique and coworkers.[24]

### Stool samples for parasite infection and microbiome analysis

Stools were collected at baseline, week 2, week 10 and week 18. Samples were analysed within 24 hours for presence of parasites. Samples were stored to create a biobank for future short-chain fatty acid and microbiome analysis.

## DATA COLLECTION AND STORAGE

Sample analysis will take place in a number of different specialist laboratories in the UK (University of Nottingham: plasma Zn, Fe and SE, water, flour), USA (NeurobioTex: nail and hair Zn concentration) and Pakistan (KMU: Comet assay, haematology, serum transferring receptor, ferritin, RBP and inflammatory markers). Comet assay tail-moment analysis will be undertaken by both Pakistan-based (KMU) and UK-based (University of Central Lancashire, UCLan) teams, and the results compared. Any significant differences will be resolved with expert support from the Children's Hospital Oakland Research Institute. Sample analysis will be undertaken

in duplicate as a minimum requirement. Data entry will be undertaken by the expert laboratory and encrypted spreadsheets sent to the data hub where a final check for anomalies will be made by the database manager. Any potential anomalies identified will be checked with the expert laboratory. All data will be managed compliant with standard BBSRC policies (http://www.bbsrc.ac.uk/web/FILES/Policies/data-sharing-policy.pdf). The UCLan will be the data hub for the project; UCLan adheres to an Open Data policy and has an open data repository. Metadata will be collected in the form of 'read me' files using basic Dublin Core. This study will generate small quantities of personal data from the analysis of blood and hair/nail samples. Most of the participants were illiterate so verbal informed consent was obtained and recorded by the project manager. Participants were given an ID number on recruitment, and this will be used in all data files. The list of names associated with the ID numbers is known only by the project manager and is stored in a locked filing cabinet. Data collected in the field on laptops are password protected. UCLan systems are automatically backed up. Transfer of data between laboratories will be in the form of encrypted Excel or Word files. Data and metadata will be securely stored in a project database and reported on publication. Data will be shared with the project steering committee during the bi-monthly meetings. This committee is composed of the PI and Co-Is. The database manager is undertaking the role of data monitoring and all members will have access to the final database.

## STATISTICAL ANALYSIS PLAN

The analysis will be performed as intention to treat. Descriptive statistics will summarise participants' characteristics and outcome measurements at each time point, using methods appropriate for the type of data. Most study outcome measurements will be quantitative data. The primary outcome, plasma Zn concentration, was collected at five time points (baseline, weeks 4, 8, 12 and 16). After adjustment for inflammatory markers and as described above, the primary outcome will be assessed in two type comparisons:

### Within-group (A and B) comparisons

Within each group of participants, the study outcome will be compared at appropriate time points (baseline will be compared with week 4 and week 8, week 4 compared with week 8, week 12 with week 16) to assess trend of the change (increase or decrease) in Zn concentrations. The pairwise comparisons will be done using paired t-test or Wilcoxon signed-rank test if normality assumptions for t-test are not satisfied. Repeated-measures ANOVA (or its non-parametric analogue) will be applied for comparisons at multiple time points (eg, to compare study outcome at baseline, week 4 and week 8).

## Comparisons between groups A and B

To answer the primary research question of determining the effectiveness of consuming flour made from agronomically biofortified wheat for improving Zn status, study outcomes will be compared between groups A and B at week 8 and week 16, complemented with comparisons at interim collection points, week 4 and week 12. The pairwise comparisons will be done using independent-samples t-test or Mann-Whitney U test as appropriate. Linear mixed modelling will be used for adjusted analyses to account for the baseline, repeated measurements and relevant covariates. Similar analysis will be done for all secondary outcomes measuring Zn status.

## ETHICS AND DISSEMINATION

Ethical approval for the research has been granted from the lead University (UCLan) (unique reference number: STEMH 697 FR) and the collaborating institution in Pakistan, Khyber Medical University. All data collected will be anonymised and data protection protocols followed when information is shared between participating institutions.

The final study methods (including any modifications) will be published in peer-reviewed journals, alongside the study outcomes on completion of the data analysis. Authorship will be restricted to those who have contributed to the design of the study, data collection and analysis, in accordance with journal guidelines. In addition to peer-reviewed journals, findings will be disseminated to the scientific community via conference presentations and abstracts. The findings of the study will be communicated to the study participants through the village elders at an appropriate community forum. More widely, the AFPK can access some of the most difficult-to-reach communities in KPK and the Federally Administered Tribal Area, including through tribal networks, local Jirga and religious leaders. We will work with our contacts in the agriculture sector to exchange knowledge with key actors. For example, the Agricultural Services Department of FFC provides soil, water, fertiliser, and plant analysis and recommendation services (in addition to other farmer education/training programmes) to the farming community throughout the country. Project partners and collaborators are well connected to ensure that new evidence can support policy.

### Author affiliations

[1]International Institute of Nutritional Sciences and Applied Food Safety Studies, Faculty of Health and Wellbeing, University of Central Lancashire, Preston, UK
[2]Institute of Basic Medical Sciences, Khyber Medical University, Peshawar, Pakistan
[3]School of Biosciences, University of Nottingham, Sutton Bonnington Campus, Leicestershire, UK
[4]Research and Development Department, Fauji Fertilizer Company Limited, Rawalpindi, Punjab, Pakistan
[5]Department of Population Health, London School of Hygiene and Tropical Medicine, London, UK
[6]Abaseen Foundation, Peshawar, Pakistan
[7]Division of Medicine, Medical Teaching Institution, Lady Reading Hospital, Peshawar, Pakistan

**Acknowledgements** We are grateful for the in-kind contribution of Fauji Fertilizer Company (FFC) Limited for crop management and production, and to HarvestPlus for supplying the Zincol-2016/NR-421 wheat grain for sowing. We would also like to acknowledge the in-kind contribution of Professor Janet King and Dr Swapna Shenvi at the Children's Hospital Oakland Research Institute (CHORI) for support with the Comet assay, and Dr Amanda Palmer and Dr Katie Healy at Johns Hopkins University for training and loan of the dark adaptation detection equipment. The authors gratefully acknowledge the Abaseen Foundation, Pakistan for use of the health centre and facilitating access to the community. We also thank Dr Svetlana Tishkovskaya at UCLan for statistical support.

**Contributors** NML, MRB, MHZ, HJMcA, MJK, EJMJ and MZA conceived the research questions and designed the study protocol. NML is the principal investigator, MRB, MJK, HJMcA, EJMJ, MHZ and MZA are co-investigators. RM is managing the community engagement and recruitment; BS, UU and GK are conducting the field work and sample collection. HO was involved in the original conception of the study and is managing the study database at the project hub (UCLan). All authors have read and approved this manuscript.

**Funding** This work is supported by BBSRC Global Challenges Research Fund, Foundation Awards for Global Agriculture and Food Systems Research (grant no. BB/P02338X/1).

**Competing interests** None declared.

**Patient consent** Obtained.

**Ethics approval** University of Central Lancashire STEMH ethics committee.

**Provenance and peer review** Not commissioned; externally peer reviewed.

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
