## [Reviewer comments · BMJ Open]

ARTICLE DETAILS

TITLE (PROVISIONAL)	Examining the effectiveness of consuming flour made from agronomically biofortified wheat (Zincol-2016/NR-421) for improving Zn status in women in a low resource setting Pakistan: Study protocol for a randomised, double blind, controlled cross over trial (BiZiFED).
AUTHORS	Lowe, Nicola; Khan, Muhammad; Broadley, Martin; Zia, Munir; McArdle, H; Joy, Edward; Ohly, Heather; Shahzad, Babar; Ullah, Ubaid; Kabana, Gul; Medhi, Rashid; Afridi, Mukhtiar

VERSION 1 – REVIEW

REVIEWER	Michael Hambidge MB BCir, FRCP ed,ScD University of Colorado School of Medicine USA
REVIEW RETURNED	28-Jan-2018

GENERAL COMMENTS	The rationale for this trial is strong and it is being undertaken in the right environment and with the ideal population. The principal investigator is recognized internationally for her leadership role in human zinc research and is very familiar with this population. The trial is ambitious especially in that it is addressing two uncertain outcomes simultaneously these being the potential of the heavy weight agricultural intervention to improve zinc status over a relatively short time period and the sensitivity of the outcome measures---the limitations of which are well recognized by the investigating team and which reflect the present status of this research field. This reviewer is left wondering if the investigators have any baseline information on plasma zinc concentrations (or other proposed outcome measures) in adult women in this population and, if so, whether mean if not every individual values are known to be low. Though these concerns are significant, in our current state of knowledge about both optimal strategies for preventing zinc deficiency and assessing zinc status, the rationale for this study remains strong. Specific comments: 1. The paper jumps from past, current, future tense sometimes appropriately and sometimes inappropriately depending on the exact status of the trial at this time. It appears that the study ins certainly underway and as it is a short term trial may well have been completed.2. Line 56: presumably the terms 'compliance and efficacy' refer to studies rather than to fortification itself?3. Though not essential to the description of the rationale, it would be interesting to know the zinc content of the trial and control wheat. Has any pilot acceptability study been undertaken on the trial
--

	wheat?Details of the nutrient, including amino acid, contents of the test and control wheat would be helpful. 4. The principal appeal of this trial is the potential of this intervention to provide a practical solution for the prevention of zinc deficiency. With this in mind, it would be relevant to include an estimate of the cost of this intensive zinc fertilization of the wheat. 5. Randomization: this seems a little fuzzy. Did you screen the households for a potentially eligible woman before including that household in the randomization? 6. To this reviewer who is not familiar with this population it would seem advantageous to have your population as homogeneous as possible esp with respect to parity, age and SES. 7. Finally, I note that the grain has already been milled which leaves me curious about storage and shelf life unless, as I hope, it has already been fed. I llk forward to learning about the outcomes in due course.
--	---

REVIEWER	Fabiana Moura U.S. Food and Drug Administration
REVIEW RETURNED	30-Jan-2018

GENERAL COMMENTS	The study protocol describes a randomized, double blind, controlled crossover trial to evaluate the effectiveness of consuming flour made from agronomically biofortified wheat on improving zinc status among low-income Pakistani women of childbearing age. It is an ongoing project, the 2-week baseline period started on October 1st, 2017. The study is well designed with two feeding periods of 8 weeks each and an adequate sample size for a crossover study design. The authors are applying cutting edge technology (DNA fragmentation) as well as the standard measure of zinc status (plasma zinc). The collection of composite meal samples for measurement of zinc and phytic acid content, and the measure the mineral content of the water from pumps in the villages are some of the strengths of the study. Other strengths are the DNA fragmentation analysis being performed by two different laboratories (UK and Pakistan) and the measurement of anti-inflammatory markers. In case the authors are not already applying a multi-pass 24-hour recall, the following manual is suggested: Rosalind Gibson, Elaine Ferguson, 2008. An interactive 24-hour recall for assessing the adequacy of iron and zinc intakes in developing countries. HarvestPlus Technical Monograph 8.
---

VERSION 1 – AUTHOR RESPONSE

Reviewer: 1

Reviewer Name: Michael Hambidge MB BCir, FRCP ed,ScD

The rationale for this trial is strong and it is being undertaken in the right environment and with the ideal population. The principal investigator is recognized internationally for her leadership role in human zinc research and is very familiar with this population. The trial is ambitious especially in that it is addressing two uncertain outcomes simultaneously these being the potential of the heavy weight agricultural intervention to improve zinc status over a relatively short time period and the sensitivity of the outcome measures---the limitations of which are well recognized by the investigating team and which reflect the present status of this research field. This reviewer is left wondering if the

investigators have any baseline information on plasma zinc concentrations (or other proposed outcome measures) in adult women in this population and, if so, whether mean if not every individual values are known to be low. Though these concerns are significant, in our current state of knowledge about both optimal strategies for preventing zinc deficiency and assessing zinc status, the rationale for this study remains strong.

Response: Thank you for your endorsement of our study rationale. We do have some data on plasma zinc concentration from a small number of women (n=15) living in this community from a previous study (unpublished). Of the 15 women, 14 had plasma zinc concentrations below 0.65 mg/L. This information, coupled with our knowledge of the local diet (vegetable based, high phytate), and the high levels of stunting in children (>40%) give us a strong rationale for believing that zinc status was likely to be low in this community.

Specific comments:

1. The paper jumps from past, current, future tense sometimes appropriately and sometimes inappropriately depending on the exact status of the trial at this time. It appears that the study is certainly underway and as it is a short term trial may well have been completed.

Response: The trial has just been completed, therefore we have amended the document to reflect this, and improve the consistency of tense.

2. Line 56: presumably the terms 'compliance and efficacy' refer to studies rather than to fortification itself?

Response: Thank you for raising this, we agree it is ambiguous. We meant the system of fortification, although it also applies to studies too. This quality assurance mechanism to ensure that fortification meets government standards in LMICs often fails, as we have experienced ourselves in our study of iodised salt in Pakistan [Lowe et al, Increasing Awareness and Use of Iodised Salt in a Marginalised Community Setting in North-West Pakistan. *Nutrients* 2015, 7(11), 9672-9682; doi:10.3390/nu7115490]. This has now been clarified in the manuscript.

3. Though not essential to the description of the rationale, it would be interesting to know the zinc content of the trial and control wheat. Has any pilot acceptability study been undertaken on the trial wheat? Details of the nutrient, including amino acid, contents of the test and control wheat would be helpful.

Response: The zinc concentration of the trial and control wheat is given in lines 177 and 178. We did conduct a pilot in summer 2016 to test the acceptability of the wheat and the protocol (Khan et al, *Proceedings of the Nutrition Society* doi:10.1017/S0029665117003457) which helped to inform the design of the final study. Unfortunately we do not have the details of the nutrient composition of the wheat yet.

4. The principal appeal of this trial is the potential of this intervention to provide a practical solution for the prevention of zinc deficiency. With this in mind, it would be relevant to include an estimate of the cost of this intensive zinc fertilization of the wheat.

Response: The health economic impact of the addition of Zn-fertilizers to the Zn concentration of high Zn wheat variety is part of the second phase of this project, starting in spring 2018. This will be published separately in due course.

5. Randomization: this seems a little fuzzy. Did you screen the households for a potentially eligible woman before including that household in the randomization?

Response: Line 137 Yes, the head of the household was approached prior to randomization to ensure that an eligible women was present in the household.

6. To this reviewer who is not familiar with this population it would seem advantageous to have your population as homogeneous as possible esp with respect to parity, age and SES.

Response: This is a homogenous, very low resource community, with all families living in basic accommodation without indoor plumbing. 90% of the families have a household income of less than 20,000 PKR per month (less than £130 per month).

7. Finally, I note that the grain has already been milled which leaves me curious about storage and shelf life unless, as I hope, it has already been fed.

Response: Line 192-193. The grain was stored whole and milled prior to distribution to the families at fortnightly intervals throughout the intervention. This has been clarified in the manuscript.

I look forward to learning about the outcomes in due course.

Response: Thank you.

Reviewer: 2

Reviewer Name: Fabiana Moura

Institution and Country: U.S. Food and Drug Administration

Please state any competing interests or state 'None declared': None declared

Please leave your comments for the authors below.

The study protocol describes a randomized, double blind, controlled crossover trial to evaluate the effectiveness of consuming flour made from agronomically biofortified wheat on improving zinc status among low-income Pakistani women of childbearing age. It is an ongoing project, the 2-week baseline period started on October 1st, 2017.

The study is well designed with two feeding periods of 8 weeks each and an adequate sample size for a crossover study design. The authors are applying cutting edge technology (DNA fragmentation) as well as the standard measure of zinc status (plasma zinc). The collection of composite meal samples for measurement of zinc and phytic acid content, and the measure the mineral content of the water from pumps in the villages are some of the strengths of the study. Other strengths are the DNA fragmentation analysis being performed by two different laboratories (UK and Pakistan) and the measurement of anti-inflammatory markers.

In case the authors are not already applying a multi-pass 24-hour recall, the following manual is suggested:

Rosalind Gibson, Elaine Ferguson, 2008. An interactive 24-hour recall for assessing the adequacy of iron and zinc intakes in developing countries. HarvestPlus Technical Monograph 8.

Response: Thank you for your suggestion. We are aware of this excellent document by Rosalind Gibson and Elaine Ferguson, and have used it to inform our 24 hour recall methodology.